# Deep Learning Combinatorial Models for Intelligent Supply Chain Demand Forecasting

**DOI:** 10.3390/biomimetics8030312

**Published:** 2023-07-15

**Authors:** Xiaoya Ma, Mengxiu Li, Jin Tong, Xiaying Feng

**Affiliations:** 1Department of Logistics Management and Engineering, Nanning Normal University, Nanninng 530023, China; mxli98@email.nnnu.edu.cn; 2Department of Economics and Management, Nanning Normal University, Nanninng 530001, China; xyfeng@email.nnnu.edu.cn

**Keywords:** demand forecasting, prediction modeling, deep learning, intelligent supply chain, new energy vehicles, SARIMA-LSTM-BP model

## Abstract

Low-carbon and environmentally friendly living boosted the market demand for new energy vehicles and promoted the development of the new energy vehicle industry. Accurate demand forecasting can provide an important decision-making basis for new energy vehicle enterprises, which is beneficial to the development of new energy vehicles. From the perspective of an intelligent supply chain, this study explored the demand forecasting of new energy vehicles, and proposed an innovative SARIMA-LSTM-BP combination model for prediction modeling. The data showed that the RMSE, MSE, and MAE values of the SARIMA-LSTM-BP combination model were 2.757, 7.603, and, 1.912, respectively, all of which are lower values than those of the single models. This study therefore, indicated that, compared with traditional econometric forecasting models and deep learning forecasting models, such as the random forest, support vector regression (SVR), long short-term memory (LSTM), and back propagation neural network (BP) models, the SARIMA-LSTM-BP combination model performed outstandingly with higher accuracy and better forecasting performance.

## 1. Introduction

According to the international energy agency, China emitted 98.94 t of CO_2_ in 2020, accounting for 30.9% of global CO_2_ emissions, with the road transport sector representing a significant source of carbon emissions [1]. As a typical representative of the transport industry, the automobile industry has a large scale, a long industrial chain, a wide range of involvement, and a high degree of internationalization [2]. Currently, fossil-fuel vehicles still dominate the automobile market, but their continued use introduced serious problems involving environmental pollution and energy consumption. The new energy vehicle industry is playing an increasingly important role in optimizing the structure of carbon emissions, especially in recent years, during which the scale and systematic development of the new energy vehicle industry made world-renowned achievements.

However, the new energy vehicle industry has not made substantial progress in market expansion [3]. The “intelligent supply chain” may contribute to changing this situation. The “intelligent supply chain” is a technology and management integrated system that combines Internet of Things technology and modern supply chain management theories, methods, and technologies, and is built within and between enterprises in order to realize the intelligence, networking, and automation of the supply chain. With the development of the traditional supply chain, the penetration of technology is increasing. Many supply chains already possess advanced technological features, such as informatization, digitization, networking, automation, and so on. As for the deep learning of new technology in the automotive supply chain, it is the result of the application of modern science and technology being used to transform the traditional automobile supply chain. The application of artificial intelligence technologies such as machine learning and deep learning played an important role in the smart transformation of the traditional supply chain. Machine learning is a multi-cross-disciplinary subject, is the core of artificial intelligence, and is the fundamental way to make computers intelligent.

Deep learning is a new research field within machine learning, and it is widely used for forecasting in the supply chain field, such as in production prediction [4], price prediction [5], demand prediction, and so on. Liu et al. [6] studied the optimal pricing problem encountered by automotive supply chain members in the context of consumer low-carbon preferences and policy implementation backgrounds. Prediction is the basis of pricing. As far as the automotive supply chain, Tan et al. [7] used statistical and econometric theory and methods in order to predict the future demand for new energy vehicles. Nonlinear fitting ability is one of the advantages of deep learning; so, using deep learning methods can improve prediction results for the automotive supply chain. Based on the response to the predicted results, the cost of the entire automotive supply chain can be reduced, and the supply chain can, thus, be optimized. Demand prediction is an important part of prediction, and represents one of the main research issues in intelligent supply chains. Its further research has significant implications for the perception, prediction, and response of production demand. Demand prediction for new energy vehicles can help new energy vehicle manufacturers accurately capture the target demand of market users. An accurate prediction of new energy vehicle demand can not only help companies at various nodes of the automotive supply chain to better formulate their production plans and developmental plans, as well as improve their competitiveness, but also help new energy vehicle manufacturers accurately grasp the target demand of market users, avoid blind production of new energy vehicles, effectively reduce inventory costs, and ensure a balance between the supply and demand of new energy vehicles. At the same time, accurate demand prediction for new energy vehicles can effectively reduce transportation costs in the new energy vehicle supply chain, shorten delivery times, and meet consumer demand to a greater extent.

In summary, in the context of the “double carbon” goal, this study starts from the perspective of intelligent supply chains, and subsequently takes new energy vehicles as the research object, and demand prediction as the research theme. It innovatively proposes a combination model of SARIMA-LSTM-BP based on the seasonal autoregressive integrated moving average (SARIMA) model, the long short-term memory (LSTM) model, and the back propagation neural network (BP) model, exploring more accurate demand predictions for new energy vehicles, and aiming to contribute to the development of the new energy vehicle industry and its supply chain.

The remainder of this paper is organized as follows: the following part is a detailed literature review, the third part introduces the research methods and experimental process used for the basis of this paper, the fourth part analyzes and discusses the experimental results, and the fifth and fifth part summarizes the conclusions of this paper and prospects for the future of the field.

## 2. Literature Review

### 2.1. Research on Intelligent Supply Chain under Low Carbon and Environmental Objectives

In the process of achieving the low-carbon goal, with the withdrawal of high-carbon enterprises, the low-carbon objectives of high-emission enterprises necessitate an increase in technological transformation and in the large number of new low-carbon industries; these current conditions led to a significant change in the structure and layout of the existing intelligent supply chain. According to M. F. Lu et al. [8], digitalization is the new feature of the current supply chain, the green concept has become a new requirement for the supply chain economy, and supply chain synergy will be further amplified. In addition, Wang [9] redefined the essential connotation and basic characteristics, identified the driving factors, further determined the implementation mechanism and construction path of the digital supply chain ecosystem under the low-carbon goal, and proposed an approach to building an efficient and scientific digital supply chain ecosystem with Chinese characteristics. Scholars such as Liu [10] believe that it is necessary to further enhance the public’s awareness and preference for green and low-carbon lifestyles, actively promote the low-carbon transformation of the supply chain, guide supply chain enterprises to transform them into low-carbon ones, optimize cooperation in order to increase the interests of all parties, and improve the efficiency of the supply chain. Han [11] and other scholars believe that green supply chain management is a specific measure by which to implement the low carbon goal at the supply chain level. In recent years, considering the objective of the low-carbon goal, intelligent supply chains have been rapidly developing and gradually moving towards data and green transformation [12]. In the context of the low carbon goal, the intelligent supply chain makes full use of the public resources provided by society to maximize resource efficiency. With the support of accurate demand forecasting, the intelligent supply chain is expected to minimize the waste of resources and optimize the environmental benefits of the supply chain. Therefore, research on demand forecasting has a far-reaching and significant implication in the development of an intelligent supply chain.

### 2.2. Demand Forecasting

Demand forecasting is the estimation of future market demand. The accuracy of the forecast directly affects a company’s production plan, inventory levels, and customer satisfaction. Demand forecasting can be divided into two categories: qualitative forecasting and quantitative forecasting. Qualitative forecasting relies solely on subjective judgments in order to assess and predict product output. Common methods include group discussion, the Delphi method, etc. Quantitative forecasting is the use of data to establish mathematical models for prediction. Currently, statistical methods such as time series models and grey forecasting models, as well as intelligent algorithms such as artificial neural networks (ANNs) and support vector machines, are commonly used in demand forecasting.

Due to the strong subjective nature of qualitative forecasting, it is often influenced by the researcher’s subjectivity. Therefore, researchers often use quantitative forecasting methods for research. For example, some scholars used the autoregressive integrated moving average (ARIMA) model to predict the recall volume of cars for auto importing companies [13]. Wang et al. [14] used the multivariate grey forecasting model to predict demand for mechanical products. When the time series fluctuates greatly and the situation is less stable, the prediction results of traditional statistical forecasting methods are not ideal, and so, more researchers introduced deep learning methods for prediction. Some scholars proposed the use of the ARIMA model or multi-layer perceptron (MLP) to predict flood flows [15,16,17]; for example, Kaya [18] used an artificial neural network to predict car sales in the Turkish region. Li et al. [19] proposed a short-term prediction model based on bi-directional long short-term memory (Bi-LSTM) for intelligent power grids. Lee et al. [20] predicted the number of damaged car parts based on a recurrent neural network. In pursuit of forecasting accuracy, demand forecast models are gradually transitioning from single-model prediction to combination-model prediction. Combination models mainly include the combination of traditional forecasting methods with each other [21], the combination of traditional methods or econometric forecasting with deep learning forecasting methods [22], and combination models based on multiple deep learning single models [23]. Scholars effectively improved the accuracy of combined forecasting models by retaining and integrating the advantages of individual models. Some researchers established a grey-neural network prediction model with which to predict China’s private car ownership [24]. Some scholars predict the need for spare automotive parts based on an improved long short-term memory model, and the prediction accuracy can be improved by improving the prediction algorithm of deep learning [25]. When predicting particulate matter content in the air, some studies used ANNs and MLP integrated models to predict air pollution [26,27]. In predicting port vessel traffic flow, Zhao et al. [28] used the SARIMA-BP model, and the combination of the two is more optimal than the SARIMA model alone when dealing with more volatile data. Scholars based a combination model on the vector autoregressive moving average and long short-term memory models for predicting the output voltage of electric vehicle batteries [29]. There are also researchers who combine deep learning methods in order to optimize the accuracy of forecasting models; for example, Hu et al. [30] predicted China’s industrial carbon peak based on the BP-LSTM model. Predicting with two deep learning models is far more accurate than predicting with a single deep learning model. Generally speaking, using a combination forecasting model can combine the advantages of different models, effectively deal with the problem of unstable time series, improve the prediction speed, and reduce errors. Currently, demand forecasting involves a wide range of research fields and rich research content, but there is a lack of research on demand forecasting for new energy vehicles, which affects the optimization and adjustment of the new energy vehicle supply chain. The related research content urgently needs to be supplemented.

### 2.3. Demand Prediction of New Energy Vehicles

In the field of new energy vehicle demand prediction, Hu et al. [31] analyzed new energy vehicle demand by using and comparing various machine learning methods, and the final experimental results showed that random forest had the best prediction ability. Tan et al. [7] used a vector autoregression (VAR) model with search indexing based on the autoregressive moving average (ARMA) model in order to predict new energy vehicle demand. Dong et al. [32] proposed a new energy vehicle scale prediction method that included nonlinear residual components and considered both linear and nonlinear components. Wang et al. [33] used an ARIMA-LSTM model, which is a combination of traditional forecasting methods and deep learning, to predict new energy vehicle demand, and the predictive results were better than those of a single model. Currently, there are relatively few studies on new energy vehicle demand prediction, and there is an urgent need for more attention and research in this area. Accurate product demand forecasting is an important basis for decision making during the subsequent production, transportation, storage, and sales activities of the intelligent supply chain within the objectives of the low-carbon goal. Accurate demand forecasting is an important support for automotive companies to expand their market shares and increase profits. Therefore, this paper proposed a SARIAM-LSTM-BP combination model based on the sales data of new energy vehicles with unstable data to forecast the demand for new energy vehicles in the future.

## 3. Research Methods

### 3.1. Models

#### 3.1.1. Seasonal Autoregressive Integrated Moving Average Model (SARIMA)

The most commonly used traditional time series models include autoregressive (AR) models, moving average (MA) models, autoregressive moving average (ARMA) models, autoregressive integrated moving average (ARIMA) models, and seasonal autoregressive integrated moving average (SARIMA) models. Among them, the ARIMA model is one of the most widely used methods for univariate time series data prediction, but it does not support time series with seasonal components. The SARIMA model is an extension of the ARIMA model, introduced to address this problem, and it is applied to the prediction of univariate data with both trends and seasonality. The expression of the SARIMA model is:(1)$$SARIMA\left(p,d,q\right)\times \left(P,D,Q\right)s$$
where p represents the autoregressive order of the trend, d represents the order of trend difference, and q represents the moving average order of the trend, all of which are the same as in the ARIMA model. Of the four seasonal factors in the SARIMA model, P represents the seasonal autoregressive order, D represents the seasonal difference order, Q represents the seasonal moving average order, and s represents the time step length of a single seasonal cycle. In this study, we established the SARIMA model according to the following steps. First, the difference order was determined using the ADF test. Thereafter, p, q, P, and Q were determined using the final differential data autocorrelation function (ACF) and the partial autocorrelation function (PACF). Finally, the best model was selected using the Akaike Information Criterion (AIC) and the fitness test based on estimated residuals [34].

#### 3.1.2. Long Short-Term Memory Network (LSTM)

Long short-term memory (LSTM) is an extension of the recurrent neural network (RNN) that can solve the problem of long-term dependency that RNN cannot handle. LSTM introduces a “gate” mechanism with which it solves the problem of gradient vanishing or explosion and, therefore, it is often used to deal with time series data prediction problems. The LSTM network is mainly composed of input gates, forget gates, output gates, cell states, and hidden states. Among them, the input gate determines the number of unit states to be sent, the forgetting gate determines the number of inputs to the current cell state, the output gate determines the cell state output to the hidden state, the cell state refers to the internal memory of the LSTM cell, and the hidden state refers to the external hidden state [35]. The structure of LSTM is shown in Figure 1, with inputs at time t consisting of $X_{t},H_{t-1},C_{t-1}$, and outputs at time t consisting of $H_{t},C_{t}$.

The unit architecture of LSTM is shown in Figure 2.

The LSTM unit receives the current stat $X_{t}$ and the tuple state of the previous time step $h_{t-1}$ through the input gate $i_{t}$, the forget gate $f_{t}$*,* and the output gate $O_{t}$. Meanwhile, the state of the memory cell $C_{t-1}$ will be input to each gate as internal information. After receiving input information, the input gate, forget gate, and output gate will perform internal operations to determine whether to activate the cell tuple. The signal of the input gate is transformed with the nonlinear function and added to the memory cell state processed by the forget gate to form a new memory cell $C_{t}$. Finally, the memory cell state $C_{t}$ forms the output $h_{t}$ of the LSTM unit through the operation of the nonlinear function and the dynamic control of the output gate. The calculation formulas for each variable are as follows [37]:(2)$$i_{t}=sigmoid[W_{hi}\times h_{t-1}+W_{xi}\times x_{t}+W_{ci}\times C_{t-1}+b_{i}$$
(3)$$f_{t}=sigmoid[W_{hf}\times h_{t-1}+W_{xf}\times x_{t}+W_{cf}\times C_{t-1}+b_{f}]$$
(4)$$O_{t}=sigmoid[W_{ho}\times h_{t-1}+W_{xo}\times x_{t}+W_{co\times }C_{t-1}+b_{o}$$
(5)$$C_{t}=f_{t}c_{t-1}+t_{t}tanh[W_{hg}\times h_{t-1}+W_{xg}\times x_{t}+b_{c}]$$
(6)$$h_{t}=O_{t}\tanh\left[C_{t}\right]\;$$

#### 3.1.3. Back Propagation Neural Network (BPNN)

The back propagation (BP) neural network is a multilayer feedforward network trained with the error back propagation algorithm and is one of the most widely used neural network models. The BP network can learn and store a large number of input-to-output mapping relationships without a need to reveal the mathematical formulas of these relationships in advance. The learning process of the BP neural network algorithm combines forward propagation and back propagation. In the forward propagation process, information is first propagated from the input layer to the hidden layer, where it is processed and then transmitted to the output layer. If the output layer does not indicate the expected output, the error signal generated is back propagated from the output layer to the input layer. In the back propagation process, the inter-layer weights are gradually adjusted to reduce the error. Figure 3 shows the neural network model structure.

The input, hidden, and output layers are the basic components of a neural network [39], connected by weight connections and using activation functions in order to calculate outputs, which mainly include the following steps:
Network initialization:

Given a sequence (x, y) provided by the system, the number of nodes in the input layer is determined and represented as I, the number of nodes in the hidden layer is represented as J, and the number of nodes in the output layer is represented as K. The link weights $W_{ij}$ and $W_{jk}$ between the input and hidden layers and the hidden and output layers, respectively, are initialized. Additionally, the biases for the hidden layer a and for the output layer b are initialized, and the learning rate and activation function are provided.
2.Hidden layer calculation:
(7)$$h_{i}=f\left(\sum_{i=1}^{I}W_{ij}\times x_{i}-a_{i}\right),j=1,2,3\ldots J$$
f is the activation function, such as sigmoid,
(8)$$f=\frac{1}{1+e^{-x}}$$3.Output layer calculation:
(9)$$y_{k}=\sum_{j=1}^{J}W_{jk}\times h_{j}-a_{j},k=1,2,3\ldots K$$4.Error calculation:
(10)$$\mu_{k}=O_{k}-y_{k},k=1,2,3\ldots K$$5.Use gradient descent to update weights and biases in order to reduce errors.

In gradient descent, the parameters $W^{\left(L\right)}$ and $b^{\left(L\right)}$ at layer L are updated as follows:(11)$$W^{\left(L\right)}\leftarrow W^{\left(L\right)}-\beta \frac{\partial f(W,b)}{\partial W^{\left(L\right)}}$$
(12)$$b^{\left(L\right)}\leftarrow b^{\left(L\right)}-\beta \frac{\partial f\left(W,b\right)}{\partial b^{\left(L\right)}}$$

### 3.2. Stacking Ensemble Method

During the research process, the research object is often influenced by various factors, and each research model has different emphases. It is difficult to grasp its complete, objective, and comprehensive development laws using only one training model with which to predict the research object. Ensemble learning can often achieve significantly superior generalization performance than a single learning model by combining multiple learners [40]. Common ensemble learning methods include boosting, bagging, and stacking. Among them, stacking more greatly emphasizes the differences between learners when performing model integration [37]. Therefore, this paper adopted the stacking ensemble learning method. First, the initial samples were experimented upon and predicted using a single model. Thereafter, the results of the SARIMA model and LSTM model, which are two primary learners, were combined to generate a new dataset. Subsequently, the values from the new dataset were used as input values in order to train the secondary learner, and a better-fitting combination model was obtained. Figure 4 shows the flowchart of the stacking ensemble model in this paper.

### 3.3. Evaluation Criteria

Various indicators are used to evaluate the sales and trend forecasting results of new energy vehicles. The mean squared error (MSE), root mean squared error (RMSE), and mean absolute error (MAE) are used to evaluate prediction errors, with lower values indicating higher prediction accuracy. The RMSE can be used to compare the advantages and disadvantages of different prediction models. The MAE does not suffer from the problem of positive and negative errors canceling each other out and, therefore, can better reflect the actual situation of prediction error.
(13)$$MSE=\frac{1}{n}\sum_{i=1}^{n}\left(y_{i}-\hat{y_{i}}\right)$$
(14)$$RMSE=\sqrt{\frac{i}{n}\sum_{i=1}^{n}\left(y_{i}-\hat{y_{l}}\right)}$$
(15)$$MAE=\frac{1}{n}\sum_{i=1}^{n}\left|y_{i}-\left.\hat{y_{i}}\right|\right.$$

## 4. Analysis and Discussion

### 4.1. Experimentation

#### 4.1.1. Experimental Data

The data used in this study were all taken from the monthly sales statistics of new energy vehicles by the China Association of Automobile Manufacturers (http://www.caam.org.cn/, accessed on 1 February 2023). The experimental data were the monthly sales data of new energy vehicles from 2014 to 2022. This study predicted future demand based on past sales data of new energy vehicles.

#### 4.1.2. Experimental Environment

In this study, the deep learning experiments were conducted in a MATLAB 2020a environment with a 64-bit Windows 10 operating system. The experiment using the seasonal difference autoregressive moving average model was conducted using EViews10 software.

#### 4.1.3. Parameter Settings

In this study, random forest, SVR, BP, LSTM, SARIMA, and SARIMA-LSTM-BP models were used to predict the demand of new energy vehicles. Among them, random forest, BP, LSTM, SVR, and SARMI-LSTM-BP were deep learning methods. After multiple rounds of experiments, Table 1 is the comparison of different parameters of SARMI-LSTM-BP model. After experimental comparison of different parameters of each model, their optimal parameters were finally obtained, as shown in Table 2.

#### 4.1.4. Experimental Analysis

In this study, random forest, BP, LSTM, SVR, SARIMA, SARIMA-LSTM-BP, and other methods were used to predict the demand of new energy vehicles. Among them, the SARIMA model is a traditional forecasting method, while the random forest, BP, LSTM, and SVR models, in addition to the SARIMA-LSTM-BP combination model, are deep learning prediction models. Based on the literature review [28,30,31], this study selected the classic SARIMA traditional forecasting models and single neural network forecasting models such as random forest, LSTM, SVR, and BP, and compared them with the innovative SARIMA-LSTM-BP combination forecasting model proposed in this study. The SARIMA-LSTM-BP combination model aims to balance the elements of time series analysis and the fitting accuracy of deep learning neural networks. As shown in Table 3, the parameter values of SARIMA, BP, and LSTM were significantly better than those of random forest and SVR, indicating that these three methods are more suitable for predicting the demand for new energy vehicles, as relevant to in this study.

As shown in Figure 5, the SARIMA-LSTM-BP model was compared with three single prediction models, namely BP, LSTM, and SARIMA, and it can be seen that the RMSE of the SARIMA-LSTM-BP model was lower than that of any single model. Based on the RMSE values, it can be concluded that the SARIMA-LSTM-BP model provided more accurate predictions than BP, LSTM, or SARIMA. 

After comparing the RMSE values of multiple models, the SARIMA, BP, LSTM, and SARIMA-LSTM-BP models were selected. This study added four evaluation metrics, including MSE, MAE, mean, and median, to these four models. Among them, MSE and MAE focus more on the performance and prediction accuracy of the models, while mean and median pay more attention to the characteristics of the dataset. These two types of evaluation metrics have different emphases when evaluating the models. MSE can evaluate the degree of data changes, and the lower the value of MSE, the more accurately the prediction model describes the experimental data. The MAE value can be used to better reflect the actual situation of prediction value errors. The mean can be used to represent the central tendency of a set of data, and it is an important indicator for describing the overall characteristics of a dataset. On the other hand, the median is a typical value of a set of data and is not influenced by extreme values.

As shown in Figure 6, both the MSE and MAE values of the SARIMA-LSTM-BP model were superior to those of the single prediction models. In this study, the SARIMA-LSTM-BP model obtained the lowest MSE and MAE values during prediction, further demonstrating its superiority among the four prediction models.

Due to the different time steps involved in different models, the number of predicted values for each model was not the same. Therefore, we selected the monthly forecast data of new energy vehicles from 2015 to 2022 for different models to ensure the fairness of the experiment. As shown in Figure 7, the mean and median of the predicted data sets for each model were very close to the mean and median of the original data. The data show that the SARIMA-LSTM-BP model had the closest mean to the mean of the raw data, proving that this model performed the best. The median was sensitive to the distribution of the data set and outliers. Since this experiment’s data set was subject to seasonal fluctuations and contained many outliers, the median fluctuation of the SARIMA-LSTM-BP model was not optimal, but within a reasonable range. Therefore, considering all indicators, it was sufficient to prove that the performance of the SARIMA-LSTM-BP model is better than other models.

Additionally, considering the RMSE value, it is recommended to use the SARIMA-LSTM-BP model for the future demand prediction of new energy vehicles.

### 4.2. Discussion

#### 4.2.1. Comparison and Evaluation of Prediction Models

With the continuous deepening of research, new prediction algorithms are introduced, and traditional prediction methods face various challenges. Based on the sales data of new energy vehicles, this study compared single prediction models with combined prediction models. According to the experimental parameters, the results obtained from a dataset trained and predicted using multiple models were superior to those obtained from single model prediction.

#### 4.2.2. Challenges

As emerging products with complex characteristics, new energy vehicles demonstrate a greatly fluctuating sales volume dataset. On this premise, it is necessary to compare the appropriate forecasting model repeatedly. When using a single model for data prediction, the matching degree between the model attributes and the data sample form needs to be considered. In the experiments on deep learning, the settings of hyperparameters such as the number of hidden neurons and hidden layers in the neural network model have an extremely significant impact on the prediction results, which are related to the success or failure of the experiment. In order to achieve the optimal prediction results, it is necessary to repeatedly debug relevant parameters in order to reduce the loss value, as well as to improve the fitting degree and experimental accuracy. Although many documents have shown that the precision of deep learning prediction models is better than that of traditional prediction methods, SARIMA, as a traditional prediction method, also showed better prediction accuracy when compared with other single deep learning prediction methods for different experimental objects and data characteristics, especially in this study. Therefore, in subsequent prediction experiments, it is not advisable to rely solely on experience to directly choose deep learning prediction and fail to consider traditional prediction methods.

#### 4.2.3. Prediction

Different deep learning prediction models also have different prediction results. In this study, LSTM and SARIMA-LSTM-BP, two prediction models with good performance, were selected to predict the sales volume of new energy vehicles for the next 12 months, as shown in Figure 8. The SARIMA-LSTM-BP model not only considered the influence of seasons on the sales of new energy vehicles, but also took into account the accurate prediction ability of the deep learning prediction model. The prediction results show that the demand for new energy vehicles will generally show an upward trend with in the next year. Among them, May and November were the peak periods of local growth. Starting in July, the predicted demand for new energy vehicles in each month of the second half of the year was above 650,000 units, and supply chain production companies can reasonably produce according to this predicted quantity and make sufficient preparations for production and sales in the second half of the year.

## 5. Summary and Prospect

### 5.1. Conclusions

From the perspective of intelligent supply chains, the research carried out and presented in this paper detailed a comparative study of multiple models for predicting the demand for new energy vehicles in China. Focusing on the demand forecasting of new energy vehicles, the authors of this paper proposed a combined forecasting model of SARIMA-LSTM-BP, which was, thereafter, compared with the traditional linear fitting forecasting method and classical deep learning single forecasting models, and discussed the influence and performance of different models for the demand forecasting of new energy vehicles. The experimental results were as follows: (1) The RMSE values of the random forest, SVR, BP neural network, SARIMA, LSTM, and SARIMA-LSTM-BP models were 18.608, 4.664, 3.850, 3.469, 3.657, and 2.757, respectively; the RMSE value of the SARIMA-LSTM-BP model was the lowest. (2) The MSE values of the BP neural network, SARIMA, LSTM, and SARIMA-LSTM-BP models were 14.823, 11.252, 13.379, and 7.603, respectively, and the MAE values were 2.550, 2.266, 2.188, and 1.912, respectively. The experimental results showed the following: (1) Comparing the validation parameters, the MSE and MAE of the SARIMA-LSTM-BP, LSTM, and SARIMA models, the combination model had the best predictive performance, followed by LSTM. (2) The SARIMA-LSTM-BP combination model paid more attention to the influence of seasonal factors on the basis of the LSTM models, and so, the SARIMA-LSTM-BP model provided more comprehensive and accurate predictions. (3) The demand forecast of new energy vehicles based on the SARIMA-LSTM-BP and LSTM models is generally on the rise, which means that the overall demand of new energy vehicles in China is increasing, and supply chain enterprises can make reasonable plans and decisions on production and supply chain operation according to the forecast data. (4) The forecast data showed that the demand for new energy vehicles will increase significantly in May and November of next year, which may be related to the summer vacation suitable for travel and the shopping carnival in November. The forecast data will become a more accurate decision-making basis for marketing and logistics planning.

Demand forecasting is helpful in order to effectively reduce the logistics production cost, improve the efficiency of the whole supply chain, weaken the “bullwhip effect”, ensure the stability of the smart supply chain, and cultivate the advantages of the smart supply chain. This study predicted the demand for new energy vehicles through five single models and the SARIMA-LSTM-BP model, and multiple experiments showed that the accuracy of the SARIMA-LSTM-BP model was superior to that of a single model.

The contributions of this paper are as follows: (1) Starting from the perspective of the intelligent supply chain, the results of this study expand the research field of demand forecasting by focusing on new energy vehicles, and supplementing the relevant research in the hot industry of new energy vehicles. (2) An innovative demand forecasting combination model, SARIMA-LSTM-BP, was proposed, and its performance was better than traditional linear prediction models and classic deep learning single models. (3) Based on the original data, future demand for new energy vehicles in the next year was predicted using two models with high accuracy, which provides a basis for planning and decision-making for the supply chain system and related enterprises in all aspects of the supply chain. (4) It will make beneficial contributions to the promotion of new energy vehicles, energy conservation and emission reduction, and, thus, it will help toward achieving the low-carbon goal.

### 5.2. Prospect

The vigorous promotion of new energy vehicles is conducive to alleviating environmental pollution and energy consumption, as well as accelerating the achievement of the low-carbon goal. This study focused on demand forecasting of new energy vehicles in the intelligent supply chain under the low-carbon goal, and proposed an innovative model, which enriched the existing research content and expanded the research field of deep learning. The vigorous promotion of new energy vehicles will alleviate environmental pollution and energy consumption, and accelerate the achievement of the low carbon goal. The demand forecasting results can provide reference information for other nodes in the supply chain, avoid waste caused by the blind production of supply chain node enterprises, accelerate the intelligent upgrade of the entire industry supply chain, and provide a strong basis for the intelligent supply chain to reduce costs and increase benefits. The author proposed a combined SARIMA-LSTM-BP model based on traditional forecasting methods and deep learning. Although this model is superior to the classical single forecasting models at this stage, its accuracy still has room for improvement. We should continue to work toward infinitely increasing the advantages of the single models included in the composite model. Currently, research is focused on constructing deep learning prediction models within the intelligent supply chain field. In the future, research can aim to improve the accuracy of demand prediction in the automotive industry and other related industries, and to further study the combination and optimization of different models as a continued research direction, providing a basis for the future of smart supply chain resource allocation.

## Figures and Tables

**Figure 1 biomimetics-08-00312-f001:**
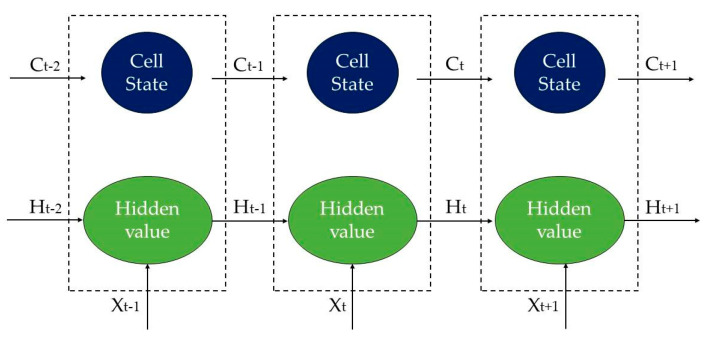
This is the structure diagram of LSTM.

**Figure 2 biomimetics-08-00312-f002:**
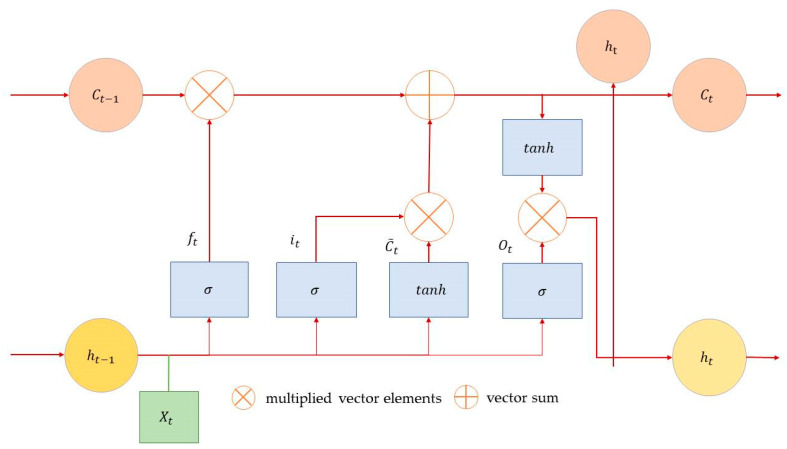
This is the cell structure of LSTM [36].

**Figure 3 biomimetics-08-00312-f003:**
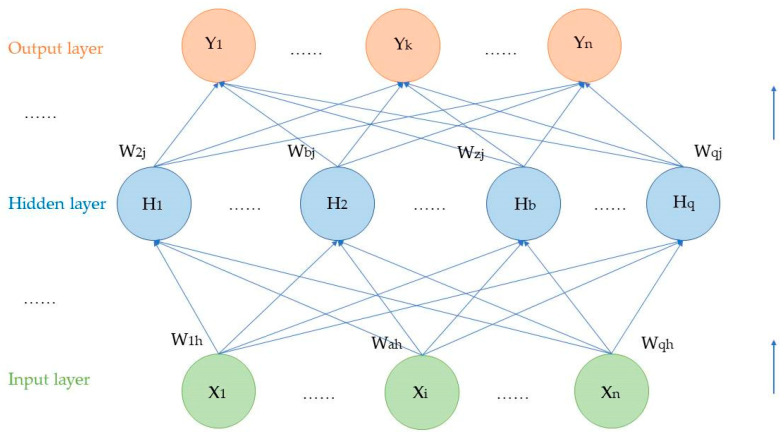
This is a neural network model structure diagram [38].

**Figure 4 biomimetics-08-00312-f004:**
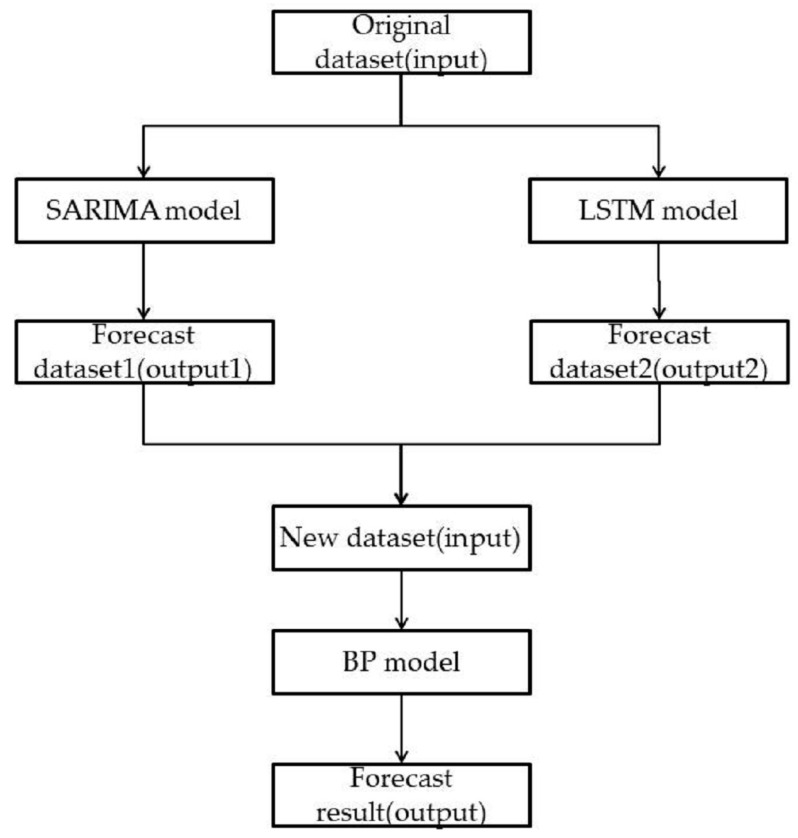
This is a flow chart of the stacking integrated model.

**Figure 5 biomimetics-08-00312-f005:**
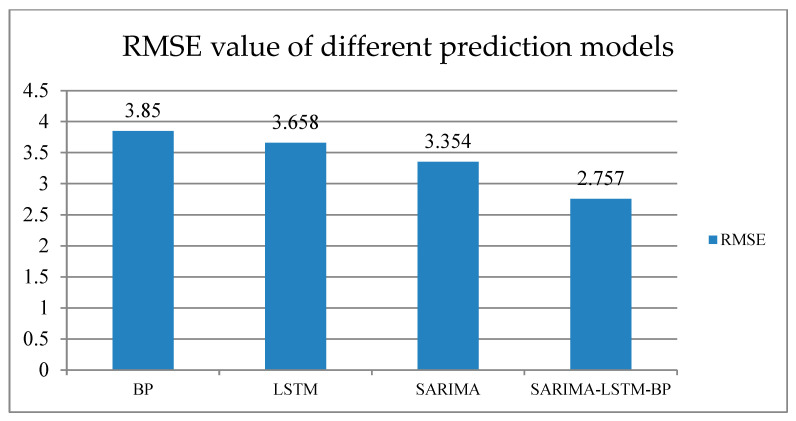
This is a comparison of the RMSE values for the different prediction models.

**Figure 6 biomimetics-08-00312-f006:**
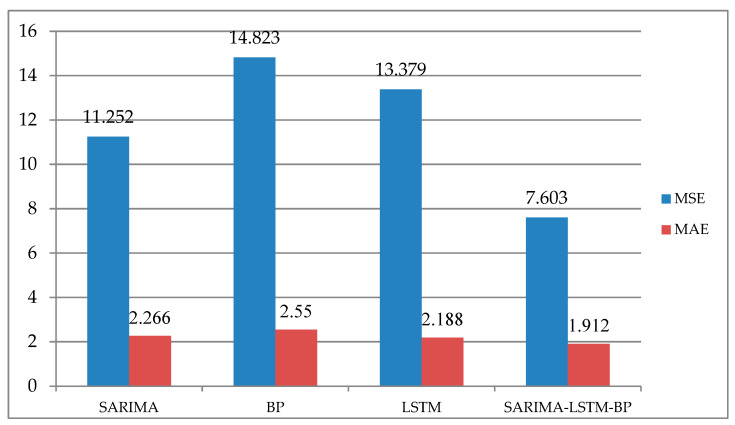
This is a comparison of the metrics of different models.

**Figure 7 biomimetics-08-00312-f007:**
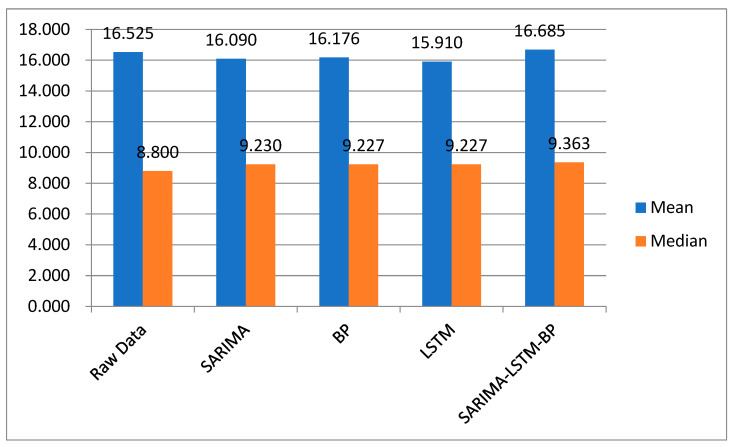
This is a comparison of the mean and median of the different models.

**Figure 8 biomimetics-08-00312-f008:**
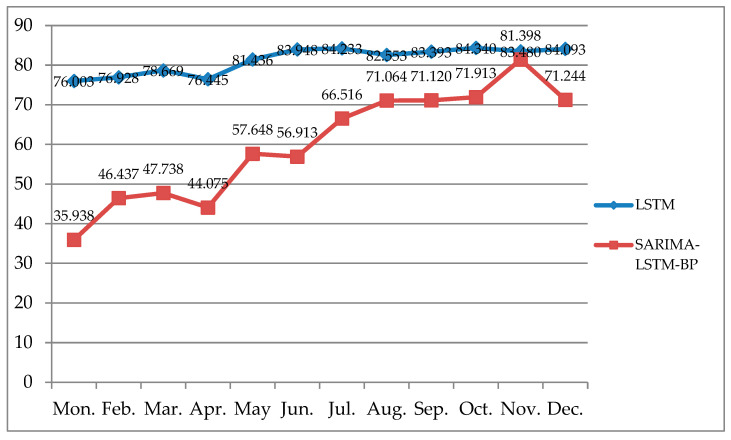
This is a comparison of the different predictions.

**Table 1 biomimetics-08-00312-t001:** This is a comparison of different parameters of the SARMIA-LSTM-BP model.

MSE	LSTM’s Hidden Layer Size Is 2	LSTM’s Hidden Layer Size Is 3	LSTM’s Hidden Layer Size Is 4
BP’s hidden layer size is 7	26.416	11.665	16.794
BP’s hidden layer size is 8	26.002	7.672	13.468
BP’s hidden layer size is 9	29.629	13.717	13.718

**Table 2 biomimetics-08-00312-t002:** This is a comparison of the parameters of different models.

Model	Model Parameter
Random Forest	Trees is 100, leaf is 5
BP	Hidden layer size is 5
LSTM	Feedback delays are 12, hidden layer size is 3
SVR	Penalty coefficient is 1, the highest degree of the kernel function is 3
SARIMA-LSTM-BP	In the LSTM, Feedback delays are 12, hidden layer size is 3; in the BP, hidden layer size is 8

**Table 3 biomimetics-08-00312-t003:** This is a comparison of the metrics of different models.

Prediction Model	SARIMA	Random Forest	SVR	BP	LSTM	SARIMA-LSTM-BP
RMSE	3.354	18.608	4.664	3.850	3.658	2.757
MSE	11.252	346.255	21.752	14.823	13.379	7.603
MAE	2.266	9.051	2.854	2.550	2.188	1.912

## Data Availability

Data are contained within the article.

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
