# Peer review of "Deep Learning Combinatorial Models for Intelligent Supply Chain Demand Forecasting"

_biomimetics, 2023, doi:10.3390/biomimetics8030312_

Round 1

Reviewer 1 Report

The paper "Deep learning combinatorial models for intelligent supply  chain demand forecasting" presents an ensemble for chain demand forecasting. The application is relevant, however, the proposal is not new in terms of modeling. In this context, some issues must be addressed, such as:

Figure 4 must be improved. The inputs and outputs of each processing module must be highlighted.

The new dataset is composed of forecasts 1 and 2 as inputs. What is the desired output for training the second module of the proposed model?

Section 3.2 is defined with a strange name.

What is the hypothesis to propose the SARIMA-LSTM-BP model? This issue is not clear.

Why are the past of the time series not used in the combination module?

The SVR model must be inserted in the experimental comparison.

The proposal must be compared to other ensemble models combined by mean and median.

Are the results statistically different?

A table with the parameters of all models used in the simulations should be inserted in the experimental protocol section.

Table 1 shows the comparison of metrics and not the comparison between parameters.

What is the paper's contribution regarding the proposed method compared to previous literature models about ensembles, such as [1, 2, 3, 4, 5]?

[1] Belotti, Jônatas, et al. "Neural-based ensembles and unorganized machines to predict streamflow series from hydroelectric plants." Energies 13.18 (2020): 4769.

[2] Cloke, Hannah L., and Florian Pappenberger. "Ensemble flood forecasting: A review." Journal of hydrology 375.3-4 (2009): 613-626.

[3] Wu, Wenyan, et al. "Ensemble flood forecasting: Current status and future opportunities." Wiley Interdisciplinary Reviews: Water 7.3 (2020): e1432.

[4] Neto, Paulo SG De Mattos, et al. "Neural-based ensembles for particulate matter forecasting." IEEE Access 9 (2021): 14470-14490.

[5] Siwek, Krzysztof, and Stanislaw Osowski. "Improving the accuracy of prediction of PM10 pollution by the wavelet transformation and an ensemble of neural predictors." Engineering Applications of Artificial Intelligence 25.6 (2012): 1246-1258.

The English language must be improved.

Author Response

Response to Reviewer

Dear Reviewer,

Thank you very much for the feedback and valuable comments. Below are the modifications and responses we have made based on your guidance.

1.Figure 4 must be improved. The inputs and outputs of each processing module must be highlighted.

 A: Thank you very much for your suggestion. According to your suggestion, we have re-examined Figure 4 and found that there is still room for improvement. We have replaced Figure 4 and marked the input and output of the data set in each module in the new figure 4.

The original figure is as:

The modified figure is as:

2.The new dataset is composed of forecasts 1 and 2 as inputs. What is the desired output for training the second module of the proposed model?

 A: Thank you very much for your valuable suggestions. The model in this paper adopts the integrated approach of stacking. The new data set consists of the predicted value 1 of the SARIMA model and the predicted value 2 of the LSTM model as inputs to the BP model. The new data set is input into the second module, the BP model, and the expected output prediction value is the final predicted value of the combined model.

3.Section 3.2 is defined with a strange name.

 A: Thanks very much for your patient advice. Based on your suggestion, we have updated the title of Section 3.2 to "Stacking Ensemble Method."

4.What is the hypothesis to propose the SARIMA-LSTM-BP model? This issue is not clear.

 A: Thank you very much for your patient advice. SARIMA has the advantage of being able to handle time series models with seasonal and periodic components. LSTM, on the other hand, introduces "gate" mechanisms that can capture trends and dependencies in sequence data, making it effective for time series prediction tasks. BP neural networks have good fault tolerance and strong nonlinear fitting capabilities. With reference to Section 2.2, most of the combined models proposed in current papers are the combined models of two single models to predict time series, such as SARMI-BP, BP-LSTM, etc. This paper hopes to give full play to the advantages of these three single models, so this paper proposes the SARMIA-LSTM-BP model, hoping to give full play to the advantages of each model and improve the accuracy of prediction results.

5.Why are the past of the time series not used in the combination module?

 A: Thank you very much for your valuable suggestions. In the SARMI-LSTM-BP model proposed in this paper, both LSTM and SARIMA models predict the future data from the past data, and output Forecast dataset1 and Forecast dataset2. Therefore, the past values of time series are used in the combined model to predict the future values.

6.The SVR model must be inserted in the experimental comparison.

 A: Thank you very much for your valuable suggestion. We have incorporated the SVR model as a comparative experiment in our study and included the evaluation metrics for SVR in Table 2. The additional content has been highlighted in blue font in the document.

7.The proposal must be compared to other ensemble models combined by mean and median.

 A: Thank you very much for your valuable suggestions and patient guidance. Based on your suggestions, we compared the average and median values among different models. However, we believe that the three evaluation metrics mentioned in the paper, MSE, MAE, and RMSE, are indicators of model performance. They are more sensitive to prediction errors and can reflect the predictive ability of the models. On the other hand, the average and median values are baseline methods that mainly focus on the characteristics of the dataset itself and cannot reflect the performance of the models. Therefore, we did not include these two indicators in our analysis.

8.Are the results statistically different?

 A: During the experiment, we observed that different models had different time steps, leading to varying numbers of predicted values for each model. For example, in the LSTM model, we used a time step of 12, resulting in 96 predicted values, while in the BP model, we used a time step of 5, resulting in 102 predicted values. Therefore, in this study, we did not employ the use of average and median values as evaluation metrics.

9.A table with the parameters of all models used in the simulations should be inserted in the experimental protocol section.

 A: Thank you very much for your valuable suggestion. We have added Section 4.1.3 Parameter Settings to the paper. This section includes a table of parameters for all models. The relevant content in the document has been highlighted in blue font.

10.Table 1 shows the comparison of metrics and not the comparison between parameters.

 A: Thank you very much for your suggestion. The title of table1 is indeed unreasonable, and we have modified the original.

The original text:

Table 1. This is a comparison of the parameters of different models.

After modification:

Table 2. This is a comparison of the metrics of different models.

11.What is the paper's contribution regarding the proposed method compared to previous literature models about ensembles, such as [1, 2, 3, 4, 5]?

 [1] Belotti, Jônatas, et al. "Neural-based ensembles and unorganized machines to predict streamflow series from hydroelectric plants." Energies 13.18 (2020): 4769.

 [2] Cloke, Hannah L., and Florian Pappenberger. "Ensemble flood forecasting: A review." Journal of hydrology 375.3-4 (2009): 613-626.

[3] Wu, Wenyan, et al. "Ensemble flood forecasting: Current status and future opportunities." Wiley Interdisciplinary Reviews: Water 7.3 (2020): e1432.

[4] Neto, Paulo SG De Mattos, et al. "Neural-based ensembles for particulate matter forecasting." IEEE Access 9 (2021): 14470-14490.

 [5] Siwek, Krzysztof, and Stanislaw Osowski. "Improving the accuracy of prediction of PM10 pollution by the wavelet transformation and an ensemble of neural predictors." Engineering Applications of Artificial Intelligence 25.6 (2012): 1246-1258.

A: Thank you very much for providing the collection of papers. We have carefully read the five articles. After comparing them, we have found that our study differs from the research subjects covered in the collection. The research subjects in the collection include flood flow and particulate matter content, while our study focuses on the demand for new energy vehicles. Typically, the choice of methods and models is determined by the different research domains. Furthermore, the predictive models proposed in our paper are different from those presented in the collection. The models in the collection mostly involve the comparison of single models or the combination of two models, whereas our paper proposes a combination model that integrates one traditional predictive model and two machine learning predictive models. Lastly, we would like to express our gratitude for providing the collection of papers. We found them inspiring and have made further improvements to our paper based on the information mentioned in the five articles. We sincerely appreciate your guidance, which has helped us enhance our research. The relevant supplemental content has been highlighted in blue font.

Author Response

Response to Reviewer

Dear Reviewer,

Thank you very much for the feedback and valuable comments. Below are the modifications and responses we have made based on your guidance.

1.I think the BP-NN you are referring to is actually the mutli-layer perceptron (MLP). It is weird to say BP compare with other deep learning architecture since LSTM is also a BP-NN.

A: Thank you for your valuable feedback. Indeed, the comparison between BP neural networks and LSTM is necessary, as they are two different types of neural network models with distinct characteristics. BP neural network (or MLP, multilayer perceptron) refers to a specific implementation of a neural network. It is a feed forward neural network composed of input, hidden, and output layers. Each neuron receives inputs from the previous layer's neurons, applies weights to them, and passes them to the next layer. BP neural networks do not have explicit memory units and are not suitable for handling sequential data. On the other hand, LSTM is a variant of recurrent neural networks (RNNs) with memory cells. LSTMs employ gate mechanisms to control the flow of information within the memory cells, enabling them to effectively handle sequential data. With input, forget, and output gates, LSTMs can capture long-term dependencies in sequences and are well-suited for time series data prediction tasks. In this study, we combine both models to leverage their respective strengths. To summarize, BP neural networks and LSTM are distinct neural network models with different architectures, training methods, and optimization focuses. BP neural networks are fed forward networks without explicit memory, while LSTMs are recurrent networks with memory cells designed for handling sequential data. The combination of both models in this study aims to harness the advantages of each model.

2.Although the authors give a comparison with MLP and LSTM, there is no comparison with other advanced time-series prediction algorithms.

A: Thank you very much for your suggestion. Based on your recommendation, we have included Support Vector Regression (SVR) as a comparative experiment in our study. SVR is a regression algorithm based on support vector machines, known for its strong modeling capability and prediction accuracy in time series forecasting. It is particularly suitable for predicting nonlinear, high-dimensional, small-sample, and non-stationary time series data, which aligns well with the characteristics of our study's original dataset, featuring small samples and non-stationary time series.

3.The ablation study is not performed to demonstrate the usage of SARIMA and LSTM. What about MLP+SARIMA?

A: Thank you for raising the question. During the background investigation phase of our research, we did come across similar research models as you mentioned. However, in order to ensure the novelty of our study, we proposed the SARIMA-LSTM-BP model. Although these models have been applied in different research domains, our approach combines them in a unique way to address the specific requirements of our study.

4.What is the size of the data? How do you guarantee there is no overfitting since your dataset is not large?

A: The data used in this study consists of monthly sales data of new energy vehicles in China from 2014 to 2022, totaling 108 data points. The experiments were conducted using MATLAB 2020a, which allowed us to generate Neural Network Training Regression plots and training performance plots. These plots were used to assess whether the models were overfitting. In this study, the scatter plot of the regression analysis shows that the data points are uniformly distributed around a straight line, and the points are close to a line with a slope of 1. Additionally, in the training performance plots, the loss functions of the SARIMA-LSTM-BP model, which is the focus of our research, consistently decrease on both the training and testing datasets, with minimal difference between them. These observations indicate that the dataset used in our study did not exhibit overfitting during the experimental process. Furthermore, the code for all the models used in this research has been packaged and submitted through the system.

5.There is no visualization of prediction samples other than these MAE/MSE numbers. What is your prediction result looks like? What does your data look like?

A: This study employed three evaluation metrics to assess the accuracy of the models, namely MAE, MSE, and RMSE. Based on existing research, these metrics are sufficient to measure the performance differences among different models. Through the screening and evaluation based on these metrics, this study selected the SARIMA-LSTM-BP model, which demonstrated good predictive performance, to forecast the future demand for new energy vehicles, as discussed in Section 4.2.3 "Prediction". The study predicted the demand for new energy vehicles for the next 12 months, and the data indicated an overall upward trend in demand. Specifically, May and November were identified as periods of local growth peaks. Starting from July, the demand for new energy vehicles remained above 650,000 units per month in the second half of the year. These findings were visualized using line graphs, as shown in Figure 6 of the article. The results of the predictions demonstrate that the proposed SARIMA-LSTM-BP model considers the influence of seasons on the demand for new energy vehicles while also leveraging the accurate forecasting capabilities of deep learning techniques.

Round 2

Reviewer 1 Report

The revised paper "Deep learning combinatorial models for intelligent supply chain demand forecasting" was improved. However, some issues must be addressed for the paper to be accepted.

Table 1 shows the optimum parameters for all analyzed models. However, the tested parameters also must be shown. This issue is crucial for the understanding of the parameters analyzed for each model.

The parameters of the models were selected based on training, validation, or testing sets?

The proposal must be compared to other ensemble models combined by mean and median. Mean and median combinations must be used to combine the forecasts SARIMA and LSTM models. These two combination functions are classical in the ensemble literature. This result is important to compare different ways to combine forecasts.

Are the results statistically different? The comparison of the models must be performed using the same test set. If the metrics are calculated from different sets, a fair comparison cannot be performed.

The authors propose applying an ensemble model for supply chain demand forecasting. In this way, the related works must be improved concerning ensemble models.

The Quality of the English Language can be improved. There are some mistakes.

Author Response

Response to Reviewer

Dear Reviewer,

Thank you very much for the feedback and valuable comments. Below are the modifications and responses we have made based on your guidance.

  1. Table 1 shows the optimum parameters for all analyzed models. However, the tested parameters also must be shown. This issue is crucial for the understanding of the parameters analyzed for each model.

A:Thank you very much for your valuable suggestions. We have added a new table in section 4.1.3 of the article to show the process of selecting the optimal parameters. For the convenience of viewing, we have marked the newly added part in green font.

  1. The parameters of the models were selected based on training, validation, or testing sets?

A: The optimization process of the model parameters is as follows: First, experiments are carried out on different parameters of the same model, and preliminary screening is conducted according to the MSE values of the training set and the test set of each experiment; then select some model parameters with good effect, compare the prediction results under different parameters of the same model, and add different model evaluation parameters such as RMSE and MAE to evaluate the model, so as to ensure that the optimal parameters of the model are selected.

  1. The proposal must be compared to other ensemble models combined by mean and median. Mean and median combinations must be used to combine the forecasts SARIMA and LSTM models. These two combination functions are classical in the ensemble literature. This result is important to compare different ways to combine forecasts.

A: Thank you very much for your advice. After much discussion, we agree on the importance of median and mean for model evaluation. Mean and median have been added as new evaluation indicators in section 4.1.4 of the article. For the convenience of checking, we will mark the newly added parts with green font.

  1. Are the results statistically different? The comparison of the models must be performed using the same test set. If the metrics are calculated from different sets, a fair comparison cannot be performed.

A: Thank you very much for your patient guidance. Due to the different time steps affecting different models, the number of predicted outputs is different. In order to ensure the fairness of the experiment, we consistently selected the monthly forecast data of new energy vehicles from 2015 to 2022, ensuring the same test set. The experimental results show that the mean of the SARIMA-LSTM-BP model is closest to the mean of the original data, proving that this model performs the best. The median is sensitive to the distribution of the data set and outliers. Since this experiment's data set is subject to seasonal fluctuations and contains many outliers, the median fluctuation of the SARIMA-LSTM-BP model is within a reasonable range, but not optimal. However, considering all indicators, it is sufficient to prove that the performance of the SARIMA-LSTM-BP model is better than other models.

  1. The Quality of the English Language can be improved. There are some mistakes.

A: Thank you very much for your valuable suggestions. We attach great importance to the submission of the paper, and have carefully corrected the errors in the paper, and hired the relevant staff of the polishing agency to help us correct the English expression. According to your suggestions, we have revised the paper again and improved the English expression quality, and the revised part has been marked with green font.

  1. The authors propose applying an ensemble model for supply chain demand forecasting. In this way, the related works must be improved concerning ensemble models.

A: Thank you for your valuable guidance. After careful study, we have decided to add relevant content about the SARIMA-LSTM-BP combination model. 1. The SARIMA model, LSTM model, and BP model can all be used for supply chain demand forecasting. Through experiments, these three individual models have shown better predictive performance than other individual models. Based on existing literature, we have innovatively proposed the SARIMA-LSTM-BP combination model, which has significantly better predictive performance than the individual models. 2. We have supplemented the optimization process of the SARIMA-LSTM-BP model parameters in the article. After comparing the predictive results of this model under different parameters, we finally selected the optimal model parameters. 3. This article also added two evaluation indicators, mean and median. Overall, the predictive results of the SARIMA-LSTM-BP model performed superiorly in the evaluation of system indicators. 4. The addition of mean and median as evaluation indicators has improved the evaluation of the dataset in the previous article. With these additions, the content related to the SARIMA-LSTM-BP model in the article has become more comprehensive. To facilitate viewing, we have highlighted the newly added content in blue font.

Reviewer 2 Report

The revised version addressed my concerns. I'd suggest accepting the current version as it is.

Author Response

Thank you very much for your guidance and affirmation, your encouragement and support for us will become our motivation to continue to work hard.